# Opportunities and Hurdles to the Adoption and Enhanced Efficacy of Feed Additives towards Pronounced Mitigation of Enteric Methane Emissions from Ruminant Livestock

**Emilio M. Ungerfeld** 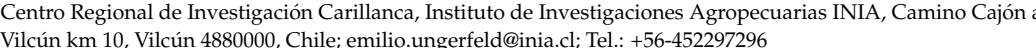

Centro Regional de Investigación Carillanca, Instituto de Investigaciones Agropecuarias INIA, Camino Cajón a Vilcún km 10, Vilcún 4880000, Chile; emilio.ungerfeld@inia.cl; Tel.: +56-452297296

**Abstract:** This paper analyzes the mitigation of enteric methane ($CH_4$) emissions from ruminants with the use of feed additives inhibiting rumen methanogenesis to limit the global temperature increase to 1.5 °C. A mathematical simulation conducted herein predicted that pronounced inhibition of rumen methanogenesis with pure chemicals or bromoform-containing algae with an efficacy higher than that obtained in most studies can be important to limiting global temperature increase by 2050 to 1.5 °C but will likely need to be accompanied by improved production efficiency and other mitigation measures. Currently, the most important limitations to the adoption of antimethanogenic feed additives are increased feeding cost without a consistent return in production efficiency and achieving sustained delivery of inhibitors to grazing animals, especially in extensive systems. Economic incentives could be applied in some countries to favor adoption of inhibitors. Changes in rumen microbial and whole animal metabolism caused by inhibiting methanogenesis could potentially be used to make the methanogenesis inhibition intervention cost-effective, although research in this direction is unlikely to yield results in the short term. Future research directions to maximize the adoption and efficacy of inhibitors of methanogenesis are examined.

**Keywords:** enteric methane; ruminants; mitigation; rumen; methanogenesis inhibition; feed additives; adoption; cost effectiveness

## 1. Enteric Methane Emissions and Climate Change

There is consensus that, in comparison to 2 °C or even higher levels of global temperature increase, limiting global temperature increase to 1.5 °C will diminish the frequency and severity of extreme climate events in the next decades [1]. Methane ($CH_4$) atmospheric concentration has doubled since industrial times and is currently second to carbon dioxide ($CO_2$) in causing global warming [2]. In addition to reaching net zero emissions of $CO_2$, achieving a strong, rapid, and sustained decrease in $CH_4$ emissions is key to rapidly limiting global warming [3]. This is largely due to $CH_4$'s relatively high global warming potential (28 times greater than $CO_2$ in a 100-year period) and relatively short life (9.25 ± 0.6 years) and perturbation time (12.4 ± 1.4 years) in the atmosphere [4]. Other benefits of decreasing $CH_4$ concentration in the atmosphere include preventing premature death due to ground-level ozone pollution and increasing crop yields [2].

As part of the overall mitigation in the emissions of greenhouse gases (GHG) to limit global warming to 1.5 °C in this century, it is estimated that global anthropogenic $CH_4$ emissions must be reduced by 40 to 45% by 2030 from 2015 levels [2]. On the other hand, past and recent trends indicate continuous growth in the emissions of $CH_4$, with a recent acceleration and projected increases in the atmospheric concentration of $CH_4$ under the current scenario [2,4–6]. Agriculture is a major source of short-term global warming through its emissions of $CH_4$ [4]. Enteric $CH_4$ emitted by domestic ruminants is the main source of agricultural $CH_4$ and accounts for about 30% of total $CH_4$ emissions from human activities [2,7]. Emissions of $CH_4$ by livestock increased by 51.4% between 1961 and 2018 [7].

The necessary decrease in enteric $CH_4$ emissions between 2020 and 2030 across various socioeconomic scenarios and climate models, compatible with a maximal 1.5 °C increase in global temperature, was estimated to be 20% on average [2]. Decreasing enteric $CH_4$ emissions is, therefore, important as part of the effort to decrease the anthropogenic emissions of GHG. The objectives of this paper are to critically examine through a mathematical simulation the possibilities of decreasing enteric $CH_4$ emissions through sustainable intensification of ruminant agriculture and the use of feed additive inhibitors of methanogenesis as the most potent strategy for enteric $CH_4$ mitigation and to analyze the opportunities and barriers to widespread adoption of inhibitors of methanogenesis for pronounced mitigation of enteric $CH_4$ emissions.

## 2. Intensification, Productivity, and Enteric Methane Emissions

Intensifying ruminant production increases the feed intake and productivity of the individual animal. Feed intake is the main driver of $CH_4$ production [8]. Increased feed intake resulting from improved feed availability and quality thus results in greater daily $CH_4$ emissions per animal. On the other hand, as animal productivity increases, a lesser proportion of dry matter intake (DMI) and of $CH_4$ emitted by an animal is associated with the animal's maintenance requirements, which has been called the "dilution of maintenance" effect. The result is a decrease in $CH_4$ emitted per unit of milk [9] or meat [10] produced or $CH_4$ intensity. There are also other animal management and feeding practices that also improve animal productivity and decrease $CH_4$ intensity, such as reducing herd size to increase individual productivity, reducing mortality and morbidity, decreasing age at slaughter, and improving fertility [11].

Improvements in production efficiency between the 2000–2004 and 2014–2018 quinquennials led to declines in $CH_4$ intensity of meat and milk from dairy cattle, buffalo, sheep, and goat protein in most regions in the world, although this was more variable for beef. Despite the decreases in $CH_4$ intensity, total $CH_4$ emitted globally by ruminants increased in the same period of time [7]. Due to the forecasted increase in production of animal products, Chang et al. [7] projected a global increase in total emissions of livestock $CH_4$ (including pigs and poultry) of between 51 and 54% by 2050 relative to 2012 assuming constant $CH_4$ intensities. With decreasing $CH_4$ intensities due to improved production efficiency following past trends, total global emissions of $CH_4$ from livestock were estimated to increase less, by 15 to 21%, between 2012 and 2050 [7]. A similar analysis for wool production in Western Australia also revealed a relationship between increased animal productivity, mostly attributed to improvements in reproductive performance, and decreased $CH_4$ intensity, along with increased total emissions of $CH_4$ [12]. Therefore, whilst production intensification and resulting improvements in animal productivity and feed efficiency can ameliorate livestock $CH_4$ emissions relative to a scenario with constant $CH_4$ intensity, total $CH_4$ emissions from livestock will likely continue rising, as a result of the increases in animal production that are, in turn, driven by the increases in human population and per capita consumption of animal products, especially in developing countries [13,14].

It has been estimated that agricultural emissions of $CH_4$ must diminish between 24 and 47% by 2050 relative to a 2010 baseline in order to contain the global temperature increase to 1.5 °C [15]. Given that the main source of agricultural $CH_4$ is livestock [16], it is reasonable to assume that enteric $CH_4$ will also need to be decreased by similar percentages between 2010 and 2050. In the same period, the consumption of bovine and ovine meat and dairy products is expected to expand by 58, 78, and 58%, respectively [13]. It follows that, in order to decrease enteric $CH_4$ emissions by 24% by 2050 relative to 2010 levels, global $CH_4$ intensity of beef, lamb, and milk production would have to decrease by 52, 57, and 52%, respectively, in relation to its 2010 levels. Likewise, decreasing enteric $CH_4$ emissions by 47% between 2010 and 2050 would require decreasing global $CH_4$ emissions intensity of beef, lamb, and milk production by 66, 70, and 66%, respectively (calculations not shown).

The same as with $CH_4$, intensifying animal production and improving animal productivity also decreases the emissions intensity of carbon dioxide equivalents ($CO_2$e; the

sum of the main three GHG $CO_2$, $CH_4$, and nitrous oxide ($N_2O$), each weighted by its heat-trapping capacity over a 100-year period), i.e., $CO_2e$ per kilogram of animal product, or carbon footprint. In some cases, decreasing the emissions of $CO_2e$ per kilogram of animal product has allowed lowering of the total number of animals sufficiently in a country or region so as to decrease the total emissions of $CO_2e$ of the livestock industries e.g., Capper et al. [9]. However, in various other cases, the decrease in the emissions of $CO_2e$ per unit of animal product occurring as a consequence of intensification was insufficient to compensate for the increase in animal production, resulting, therefore, in increased total $CO_2e$ emissions from milk and beef production [17]. Whilst producing meat and milk with a lower carbon footprint is an important goal, intensification of animal production alone is unlikely to stop the increase in total emissions of GHG from ruminant production, much less mitigate them. Specific additional measures to ameliorate the emissions of $CH_4$ and other GHG from the livestock industry are also needed.

## 3. Mitigation of Enteric Methane Emissions

It is challenging to reconcile the objectives of decreasing total emissions of enteric $CH_4$ from ruminant production and at the same time increase the global supply of animal products. Therefore, several strategies to mitigate enteric $CH_4$ emissions from ruminants are being investigated: increasing feed efficiency, genetic selection of animals with lower $CH_4$ production, modifying diet formulation and concentrate and forage processing, grazing management, the addition of oils to the diet, use of chemical inhibitors of methanogenesis, dietary inclusion of algae with antimethanogenic compounds, alternative electron acceptors, phytocompounds, defaunation (elimination of rumen protozoa), immunization against methanogens, early-life interventions, and archaeal phages, among others. For more information, readers are referred to various excellent published reviews [18–25].

The effectiveness of all [26–28] or some [29–34] enteric $CH_4$ mitigation strategies currently available has been quantified through various meta-analyses. In their meta-analysis, Arndt et al. [27] identified increasing the individual animal feed intake (by, on average, 58%, without altering the composition of the diet) as the most effective strategy to decrease the $CH_4$ intensity of milk production (by, on average, 16.7%), while simultaneously increasing animal productivity. Secondly, they identified the utilization of inhibitors of methanogenesis (including 3-nitrooxypropanol (3-NOP) and bromoform-containing red algae *Asparagopsis* spp.) as the most effective strategy to decrease total daily emissions of $CH_4$ per animal (by, on average, 35.2%) and emissions of $CH_4$ per kilogram of milk produced (by, on average, 31.8%), without negatively affecting animal productivity [27].

Using the average decreases in $CH_4$ production found in their meta-analysis, Arndt et al. [27] estimated that the adoption of increased feed intake or inhibitors of methanogenesis, or both antimethanogenic measures in combination, could allow containing of global temperature increase by 1.5 °C by 2030 but not by 2050, even if applied under an unrealistic scenario of global 100% adoption [27]. The conclusions of the analysis by Arndt et al. [27] illustrate the challenges and difficulties of increasing ruminant production while decreasing the emissions of enteric $CH_4$ and $CO_2e$.

## 4. Projection of Global Enteric Methane Emission under Different Scenarios of Intensification and Adoption of Inhibitors of Methanogenesis

A projection of emission of enteric $CH_4$ from beef, lamb, and bovine milk production and the sum of the three products between present time and 2050 was herein conducted, combining scenarios of production intensification and adoption of feed additives inhibiting rumen methanogenesis. The evolution of total global emissions of enteric $CH_4$ was simulated considering future increases in global production of beef, lamb, and bovine milk and, depending on each scenario of production intensification, future decreases in $CH_4$ intensity of beef, lamb, and bovine milk production.

The analysis considered two different scenarios for intensification of production: (i) constant production efficiency, assuming constant 2016 levels of $CH_4$ emission intensities

of beef, lamb, and bovine milk, as reported by Chang et al. [7] for the 2014–2018 quinquennial period and (ii) improved production efficiency, assuming decreasing $CH_4$ emission intensities of beef, lamb, and bovine milk at constant rates. Constant annual rates of decline in $CH_4$ intensity of beef, lamb, and milk production were calculated from changes in global $CH_4$ intensities reported for those industries by Chang et al. [7] between the 2000–2004 and 2014–2018 quinquennials.

Chang et al. [7] estimated $CH_4$ intensities using three different methods; an average of the values reported with the three estimation methods was considered for $CH_4$ intensities of global lamb and milk production. As for beef production, Chang et al. [7] reported an increase in $CH_4$ intensity between the 2000–2004 and 2014–2018 quinquennials with one of their methods of estimation and decreases with the other two, with increasing $CH_4$ intensity on average. Assuming that the global $CH_4$ intensity of beef production between 2010 and 2050 is likely to decrease as beef production increases [10,35], only the two estimations reporting declining $CH_4$ intensity of global beef production were averaged for the projection herein conducted. Annual constant rates of decline in $CH_4$ intensity were then calculated between 2002 (corresponding to the 2000–2004 quinquennial period) and 2016 (corresponding to the 2014–2018 quinquennial period; Table 1).

**Table 1.** Estimated rates of change in production and methane ($CH_4$) intensity of beef, lamb, and milk between 2010 and 2050.

| Animal Product | Metric | Period Used for Estimation | Total Change (%) | Rate (%/yr) | Source Used for Estimation |
|---|---|---|---|---|---|
| Beef | Production | 2010–2050 | 58 | 1.15 | FAO [13] |
| Lamb | Production | 2010–2050 | 78 | 1.45 | FAO [13] |
| Milk | Production | 2010–2050 | 58 | 1.15 | FAO [13] |
| Beef | $CH_4$ intensity | 2002–2016 | −0.52 | −0.037 | Chang et al. [7] [1] |
| Lamb | $CH_4$ intensity | 2002–2016 | −7.27 | −0.54 | Chang et al. [7] [2] |
| Milk | $CH_4$ intensity | 2002–2016 | −9.55 | −0.71 | Chang et al. [7] [2] |

[1] Average of the two negative estimates. [2] Average of three estimates.

Both scenarios of production efficiency (constant or declining $CH_4$ intensity) were combined with four different scenarios of adoption of feed additive inhibitors of methanogenesis as the most potent strategy available to mitigate total emissions of enteric $CH_4$ per animal [26,27]: 0, 25, 50, or 100% as the theoretical maximum worldwide adoption of feed additive inhibitors of methanogenesis in global production of beef, lamb, bovine milk, and all three combined. Percentages of global adoption refer to the percentage of global beef, lamb, and bovine milk produced with the use of inhibitors of methanogenesis, rather than to the percentage of total animals receiving the additives. The adoption of inhibitors of methanogenesis was simulated to take place gradually within a five-year period beginning in 2023 and as a linear function of time until reaching the maximum for each scenario of adoption.

The $CH_4$ intensity of beef, lamb, or bovine milk in each year and scenario was multiplied by the global production of the corresponding animal product to obtain the total $CH_4$ emissions associated to each product. Initially, the use of inhibitors of methanogenesis in milk production was modeled as causing a 31.8% decrease in $CH_4$ intensity, as reported by Arndt et al. [27]. Arndt et al. [27] did not report an effect of inhibitors of methanogenesis on $CH_4$ intensity of meat production. For meat production, a 26.6% decrease in $CH_4$ intensity of beef and lamb production, as the average of the range (−13.2 to −39.9%) for the effect of 3-NOP reported by Almeida et al. [26] for $CH_4$ intensity of milk and body mass gain, was initially considered for the analysis. A stronger effect of methanogenesis inhibitors in dairy than in beef animals agrees with a meta-analysis conducted for 3-NOP [29]. Calculations of total $CH_4$ emissions were conducted based on decreases in $CH_4$ intensity caused

by inhibitors of methanogenesis rather than decreases in daily $CH_4$ emitted per animal, because total $CH_4$ emissions can be calculated from the forecasted increases in total output of animal products and changes in $CH_4$ intensity, as opposed to using decreases in daily $CH_4$ per animal for calculating total $CH_4$, which would require predicting future changes in animal numbers.

From projected increases in production of beef, lamb, and milk of 58, 78, and 58%, respectively, between 2010 and 2050 [13], constant yearly rates of increase of 1.15, 1.45, and 1.15% for beef, lamb, and milk production, respectively, were calculated (Table 1). Given that Chang et al. [7] reported $CH_4$ intensity as kg $CH_4$ per kg of animal protein (of beef, lamb, or bovine milk), the future global production of each animal product was also calculated as kilograms of protein according to FAO [36].

Figures 1–3 depict the predicted evolution between present time and 2050 of enteric $CH_4$ emissions for different scenarios in which 0, 25, 50, or 100% of beef, lamb, and milk, or the sum of all three products (Figure 4) is produced with the use of inhibitors of rumen methanogenesis. The evolution of enteric $CH_4$ production for each scenario of adoption of inhibitors of methanogenesis was simulated under constant 2016 levels of $CH_4$ intensity or under decreasing $CH_4$ intensities, according to the rates in Table 1. The upper and lower targets of a 24 and 47% decrease in enteric $CH_4$ emissions by 2050 relative to 2010 levels, as required to maintain the global temperature increase within 1.5 °C [15], are shown as reference. The database used for the simulations is available [37].

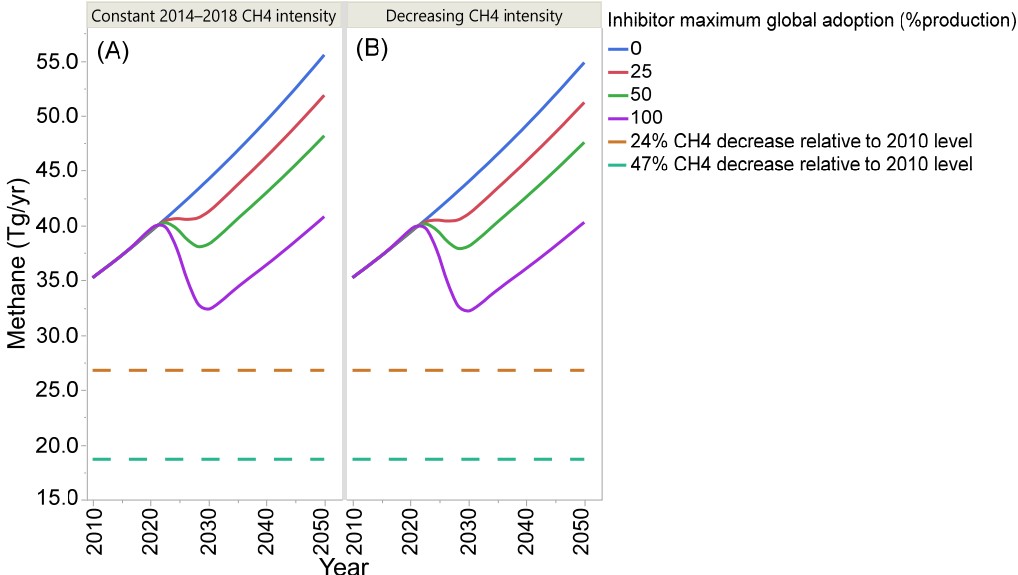

**Figure 1.** Predicted evolution of enteric methane ($CH_4$) emissions from global beef production between 2010 and 2050 considering (**A**) constant $CH_4$ intensity and (**B**) declining $CH_4$ intensity according to Table 1. Adoption of feed additive inhibitors of methanogenesis decreasing $CH_4$ intensity by 26.6% is simulated to occur at 0, 25, 50, or 100% of global beef production. Decreases in enteric $CH_4$ emissions of 24 and 47% relative to 2010 required to maintain a global temperature increase within 1.5 °C according to different socioeconomic scenarios and climate models [15] are shown in dashed lines.

Without introducing inhibitors of methanogenesis, the decrease in $CH_4$ intensity resulting from improving production efficiency can ameliorate $CH_4$ emissions from beef, lamb, bovine milk, and the sum of all three products by 0.70 (−1.3%), 1.58 (−17%), 7.80 (22%), and 10.1 (−10%) Tg/yr in 2050, compared to a scenario with constant $CH_4$ intensity. However, $CH_4$ emissions would still increase by 19.6 (+56%), 2.38 (+44%), 4.43 (+18.6), and 26.4 (+40.9%) Tg/yr from 2010 levels, in the same order, and would be 28.1 (+105%), 3.69 (+89%), 10.2 (+56%), and 41.9 (+85%) Tg/yr, in the same order, higher than the lower quartile $CH_4$ mitigation required to limit global temperature increase to 1.5 °C.

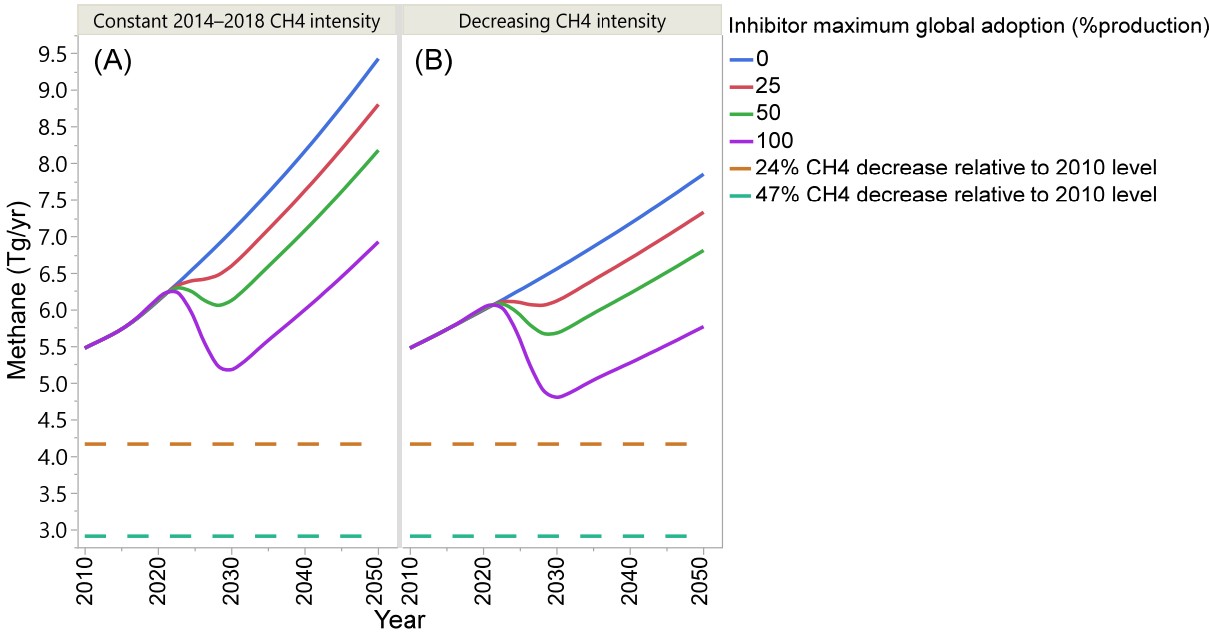

**Figure 2.** Predicted evolution of enteric methane (CH₄) emissions from global lamb production between 2010 and 2050 considering (**A**) constant CH₄ intensity and (**B**) declining CH₄ intensity according to Table 1. Adoption of feed additive inhibitors of methanogenesis decreasing CH₄ intensity by 26.6% is simulated to occur at 0, 25, 50, or 100% of global lamb production. Decreases in enteric CH₄ emissions of 24 and 47% relative to 2010 required to maintain a global temperature increase within 1.5 °C according to different socioeconomic scenarios and climate models [15] are shown in dashed lines.

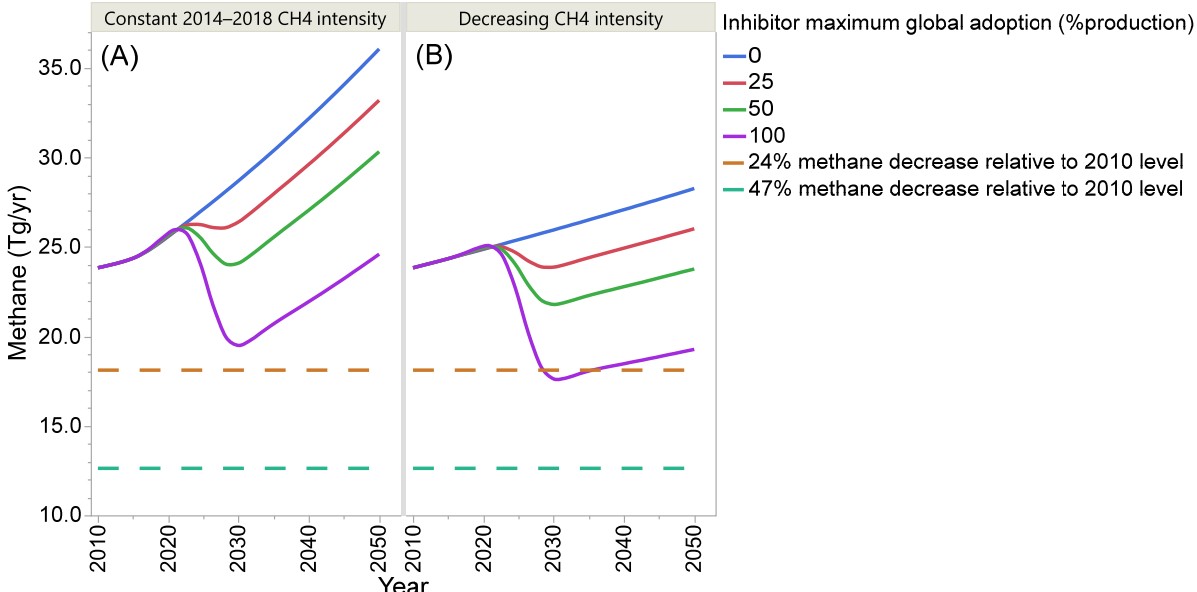

**Figure 3.** Predicted evolution of enteric methane (CH₄) emissions from global bovine milk production between 2010 and 2050 considering (**A**) constant CH₄ intensity and (**B**) declining CH₄ intensity according to Table 1. Adoption of feed additive inhibitors of methanogenesis decreasing CH₄ intensity by 31.8% is simulated to occur at 0, 25, 50, or 100% of global lamb production. Decreases in enteric CH₄ emissions of 24 and 47% relative to 2010 required to maintain a global temperature increase within 1.5 °C according to different socioeconomic scenarios and climate models [15] are shown in dashed lines.

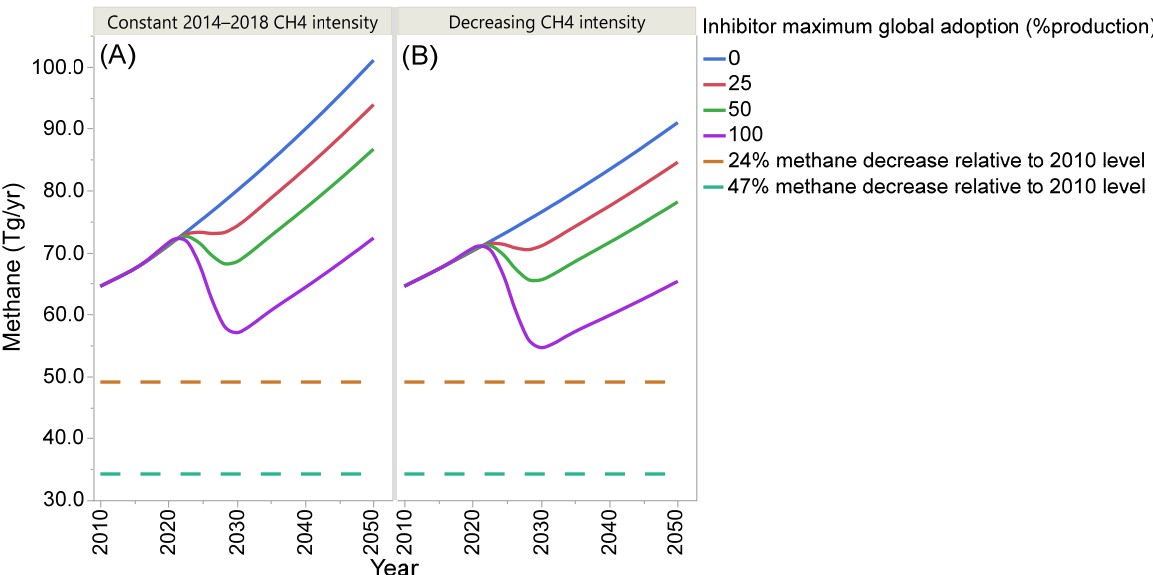

**Figure 4.** Predicted evolution of enteric methane ($CH_4$) emissions from global beef, lamb, and bovine milk production between 2010 and 2050 considering (**A**) constant $CH_4$ intensity and (**B**) declining $CH_4$ intensity according to Table 1. Adoption of feed additive inhibitors of methanogenesis decreasing $CH_4$ intensity by 26.6% for beef and lamb and by 31.8% for milk production is simulated to occur at 0, 25, 50, or 100% of global production. Decreases in enteric $CH_4$ emissions of 24 and 47% relative to 2010 required to maintain a global temperature increase within 1.5 °C according to different socioeconomic scenarios and climate models [15] are shown in dashed lines.

In this simulation, the mitigation of $CH_4$ emissions resulting from gains in production efficiency and resulting decreasing $CH_4$ intensity was estimated as minimal for beef production. The reason for this result is that the decline in $CH_4$ intensity between the 2000–2004 and 2014–2018 quinquennials for beef production estimated by Chang et al. (2021) [7] (Table 1), which was extrapolated to the 2023–2050 period in this simulation, was rather small. In fact, an increase in beef $CH_4$ intensity would have been considered had the calculation been conducted using the average of $CH_4$ intensity of the three estimations reported by Chang et al. (2021) [7] using three different methods. It is reasonable to think that the 2000–2004 to 2014–2018 tendency might change and greater gains in beef production efficiency and decreases in $CH_4$ intensity may result in the future; however, considering the results with lamb and bovine milk production, it appears likely that total $CH_4$ emissions from global beef production would still increase towards 2050, even under more optimistic improvements in $CH_4$ intensity. The adoption of practices that improve production efficiency is important to decrease the emissions of enteric $CH_4$ relative to a scenario with no improvements in $CH_4$ intensity; however, the emissions of enteric $CH_4$ would likely continue to grow if improvements in production efficiency are not accompanied by other measures.

That said, no projected scenario of enteric $CH_4$ emissions from beef or lamb production in this simulation would provide the required amelioration, even under an unrealistic 100% global adoption of inhibitors of methanogenesis (Figures 1 and 2). With 100% worldwide adoption of methanogenesis inhibitors, enteric $CH_4$ emissions from milk production would briefly decrease slightly below the upper 24% amelioration target around 2030 but would still not meet that target by 2050 (Figure 3). The same as beef and lamb production, total enteric $CH_4$ emissions from the sum of beef, lamb, and milk production was not projected to meet the minimum 24% decrease in enteric $CH_4$ emissions at any point in time between 2023 and 2050, even with a theoretical 100% worldwide adoption of inhibitors of methanogenesis (Figure 4).

## 5. Pronounced Inhibition of Rumen Methanogenesis with Feed Additives

Importantly, the use of feed additives inhibiting methanogenesis has allowed, in some studies, for considerably greater mitigation of enteric $CH_4$ production in comparison with the averages for methanogenesis inhibitors reported in the meta-analyses by Veneman et al. [28], Almeida et al. [26], and Arndt et al. [27]. Whilst, in the vast majority of studies evaluating inhibitors of methanogenesis, the extent of $CH_4$ decrease could be defined as moderate (i.e., ~30%), there have been various studies in which the inhibition of $CH_4$ production was considerably more profound (i.e., ~80% or more) [17]. Tables 2 and 3 summarize experiments in which the utilization of chemical inhibitors of methanogenesis or *Asparagopsis* spp. allowed decreasing $CH_4$ intensity of growth and fattening (Table 2) and milk production (Table 3) by 60% or more. There are studies not listed in Table 2 in which rumen methanogenesis was inhibited by 60% or more, but the animal performance was not reported and, thus, effects of methanogenesis inhibition on $CH_4$ intensity are not calculable (and, thus, cannot be used to calculate total emissions of $CH_4$ without knowing future changes in the number of animals); a more comprehensive list of experiments including treatments with inhibition of enteric $CH_4$ production equal to or greater than 60% is presented in Supplementary Table S1, demonstrating that pronounced inhibition of methanogenesis is possible. It is worth noting that, in 21 growth and fattening or maintenance experiments but in only two milk production experiments, a decrease of 60% or more in $CH_4$ production was obtained (Table S1), highlighting the need to investigate promoting pronounced inhibition of rumen methanogenesis in dairy cows.

**Table 2.** In vivo growth and fattening experimental treatments resulting in 60% or greater decrease in enteric methane ($CH_4$) production [1].

| Reference | Animal, Diet | Inhibitor/Algae (g/kg Diet DM [2]) | Experimental Period (d) | Inhibition Relative to Control Treatment (% Decrease in $CH_4$ Animal$^{-1}$ d$^{-1}$) | Inhibition Relative to Control Treatment (% Decrease in $CH_4$ kg ADG$^{-1}$) | Performance | | |
|---|---|---|---|---|---|---|---|---|
| | | | | | | DMI | ADG | G:F |
| Trei et al. [38] | Lambs, mixed | 2, 2, 2-trichloroacetamide (0.080) | 90 | 67 [3] | 67 [4] | NS [5] | NS | ↑ |
| Johnson et al. [39] | Steers, mixed | BCM (0.50) | 28 | ~65 [6] | ~68 [4] | NS | NS | - |
| Davies et al. [40] | Calves, mixed | ICI 13409 (0.20) | 196 | 63 [3] | 66 [4] | ↓ | ↑ | ↑ |
| Romero-Perez et al. [41] | Heifers, mixed | 3-NOP (0.28) | 112 | 59 | 60 | R | NS | NS |
| Vyas et al. [42], finishing diet | Steers, high concentrate | 3-NOP (0.2) | 105 | 84 | 83 | ↓ | ↓ | NS |
| Kinley et al. [43] | Steers, high concentrate | *Asparagopsis taxiformis* (1.8) | 90 | 98 | 98 | NS | ↑ | NS |
| Roque et al. [44] | Steers, high concentrate | *Asparagopsis taxiformis* (4.7) | 63 | 82 | 83 | ↓ | NS | NS |
| Alemu et al. [45] | Steers, high concentrate | 3-NOP (0.108) | 112 | 77 | 76 [4] | ↓ [7] | ↓ [7] | ↑ [7] |
| Cristobal-Carballo et al. [46] | Calves, milk replacer, concentrate, partial mixed ration, pasture | Chloroform (0.050) plus 9, 10-anthraquinone (0.50) | 84 | 90 [3] | 90 | NS | NS | - |

[1] Only experiments reporting effects on performance and/or $CH_4$ intensity are presented. For a more complete list of experiments with at least one treatment with inhibition of $CH_4$ production equal to or greater than 60%, please refer to Table S1. [2] Abbreviations: 3-NOP = 3-nitrooxypropanol; ADG = average daily body mass gain; BCM = bromochloromethane; $CH_4$ = methane; DM = dry matter; DMI = dry matter intake; G:F = body mass gain per kilogram of dry matter intake; ICI 13409 = 2,4-bis(trichloromethy1)-benzo(1,3)dioxin-6-carboxylic acid. [3] Methane concentration, rather than production, was measured by rumenotomy [38,47] or air expelled in a hood [40]. [4] Calculated from reported results. [5] NS = effects reported as not significant ($p > 0.05$); ↑ = increase ($p < 0.05$); ↓ = decrease ($p < 0.05$); ↑ = tendency to increase ($0.05 \leq p < 0.10$); ↓ = tendency to decrease ($0.05 \leq p < 0.10$); - = not reported. [6] Daily average, estimated from results shown in a graph. [7] K. Beauchemin, pers. comm.

Modifying the previous analysis of prediction of enteric $CH_4$ emissions of Section 4, by considering an average efficacy of inhibitors of methanogenesis of 60% decrease in $CH_4$ intensity of beef, lamb, and milk production, the enteric $CH_4$ amelioration required to maintain global temperature increase within 1.5 °C by 2050 could be met only if inhibitors of methanogenesis were employed in nearly 85% of beef production (Figure 5), ~75% of lamb production (Figure 6), and ~60% of milk production (Figure 7). Reaching a 24% decrease in combined enteric $CH_4$ emissions from beef, lamb, and milk production by 2050 would require a worldwide adoption slightly greater than 75% of inhibitors of methanogenesis, decreasing $CH_4$ intensity by 60% (Figure 8). Evidently, these rates of adoption are highly

unlikely. A more feasible scenario of 25% adoption of inhibitors of methanogenesis (e.g., in the most intensive production systems) with 60% efficacy would still decrease enteric $CH_4$ emissions of beef ($-8.23$ Tg/yr), lamb ($-1.18$ Tg/yr), and bovine milk ($-4.24$ Tg/yr) production by 15% relative to a scenario with no inhibitors of methanogenesis.

**Table 3.** In vivo milk production experimental treatments resulting in 60% or greater decrease in enteric methane ($CH_4$) production [1].

| Reference | Animal, Diet | Inhibitor/Algae (g/kg Diet DM [2]) | Experimental Period (d) | Inhibition Relative to Control Treatment (% Decrease in $CH_4$ Animal$^{-1}$ d$^{-1}$) | Inhibition Relative to Control Treatment (% Decrease in $CH_4$ kg FPCM$^{-1}$) | Performance | | |
|---|---|---|---|---|---|---|---|---|
| | | | | | | DMI | MY | MY:F |
| Haisan et al. [48] | Cows, mixed | 3-NOP (0.13) | 28 | 60 | 61 [3] | NS [4] | NS | NS |
| Roque et al. [49] | Cows, mixed | *Asparagopsis armata* (10) | 21 | 67 | 61 [3] | ↓ | ↓ | - |

[1] Only experiments reporting effects on performance and/or $CH_4$ intensity are presented. For a more complete list of experiments with at least one treatment with inhibition of $CH_4$ production equal to or greater than 60%, please refer to Table S1. [2] Abbreviations: 3-NOP = 3-nitrooxypropanol; $CH_4$ = methane; DM = dry matter; DMI = dry matter intake; MY = milk yield; MY:F = milk yield per kilogram of dry matter intake. [3] Calculated from reported results. [4] NS = no significant ($p > 0.05$) effects; ↓ = decrease ($p < 0.05$); - = not reported.

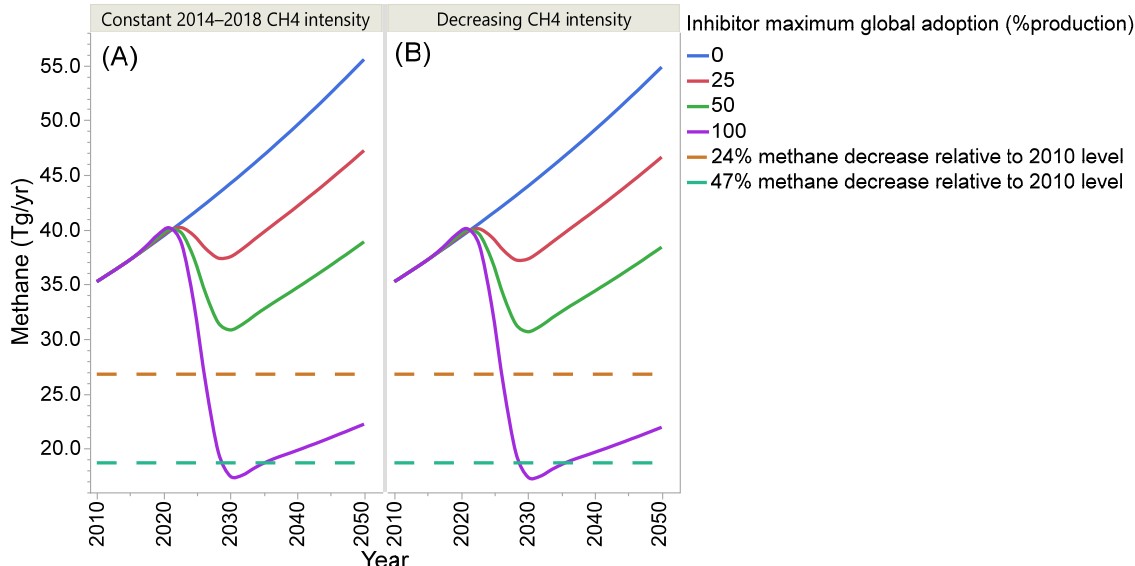

**Figure 5.** Predicted evolution of enteric methane ($CH_4$) emissions from global beef production between 2010 and 2050 considering (**A**) constant $CH_4$ intensity and (**B**) declining $CH_4$ intensity according to Table 1. Adoption of feed additive inhibitors of methanogenesis decreasing $CH_4$ intensity by 60% is simulated to occur at 0, 25, 50, or 100% of global beef production. Decreases in enteric $CH_4$ emissions of 24 and 47% relative to 2010 required to maintain a global temperature increase within 1.5 °C according to different socioeconomic scenarios and climate models [15] are shown in dashed lines.

It is thought that maximizing the use of inhibitors of methanogenesis can be an important component of a multifactorial effort to mitigate the emissions of enteric $CH_4$ from ruminant production systems. The rest of this document examines barriers to the adoption and enhanced efficiency of inhibitors of methanogenesis in ruminant production, possible solutions, and knowledge gaps and research hypotheses that can potentially generate knowledge to remove those barriers. Aspects pertaining to the adoption of inhibitors of methanogenesis that are discussed in the second part of this document are cost effectiveness and co-benefits (Section 6), delivery to animals and effectiveness in grazing systems (Section 7), and safety for animals, consumers, and the environment (Section 8). Research concerning the efficacy of inhibitors of methanogenesis is presented and discussed in Section 9.

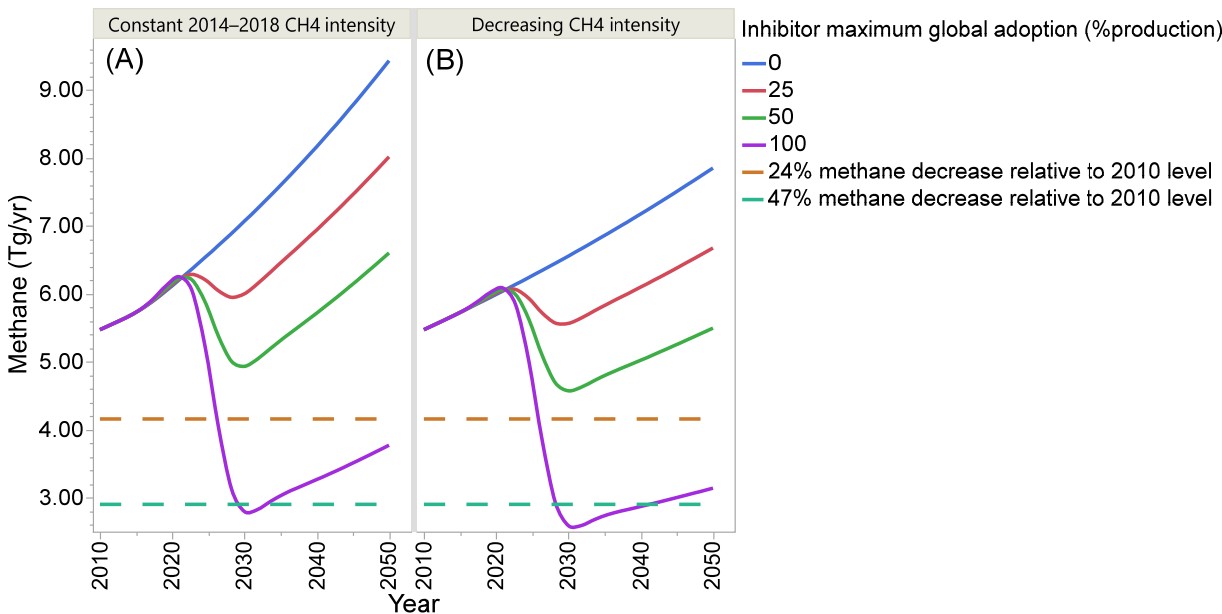

**Figure 6.** Predicted evolution of enteric methane (CH$_4$) emissions from global lamb production between 2010 and 2050 considering (**A**) constant CH$_4$ intensity and (**B**) declining CH$_4$ intensity according to Table 1. Adoption of feed additive inhibitors of methanogenesis decreasing CH$_4$ intensity by 60% is simulated to occur at 0, 25, 50, or 100% of global lamb production. Decreases in enteric CH$_4$ emissions of 24 and 47% relative to 2010 required to maintain a global temperature increase within 1.5 °C according to different socioeconomic scenarios and climate models [15] are shown in dashed lines.

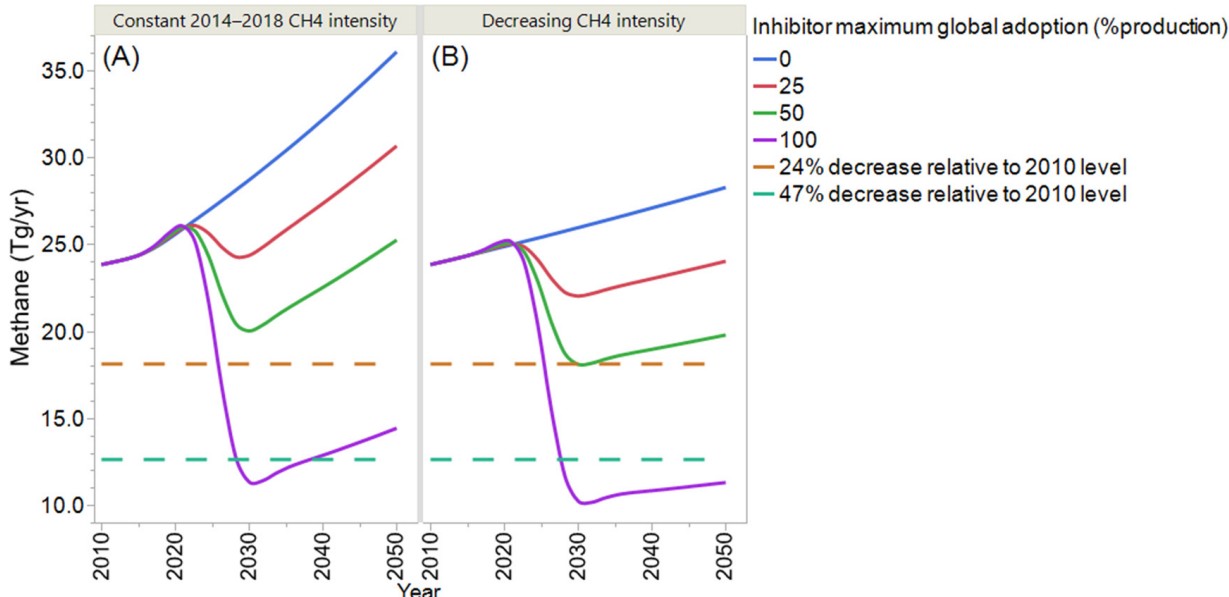

**Figure 7.** Predicted evolution of enteric methane (CH$_4$) emissions from global bovine milk production between 2010 and 2050 considering (**A**) constant CH$_4$ intensity and (**B**) declining CH$_4$ intensity according to Table 1. Adoption of feed additive inhibitors of methanogenesis decreasing CH$_4$ intensity by 60% is simulated to occur at 0, 25, 50, or 100% of global lamb production. Decreases in enteric CH$_4$ emissions of 24 and 47% relative to 2010 required to maintain a global temperature increase within 1.5 °C according to different socioeconomic scenarios and climate models [15] are shown in dashed lines.

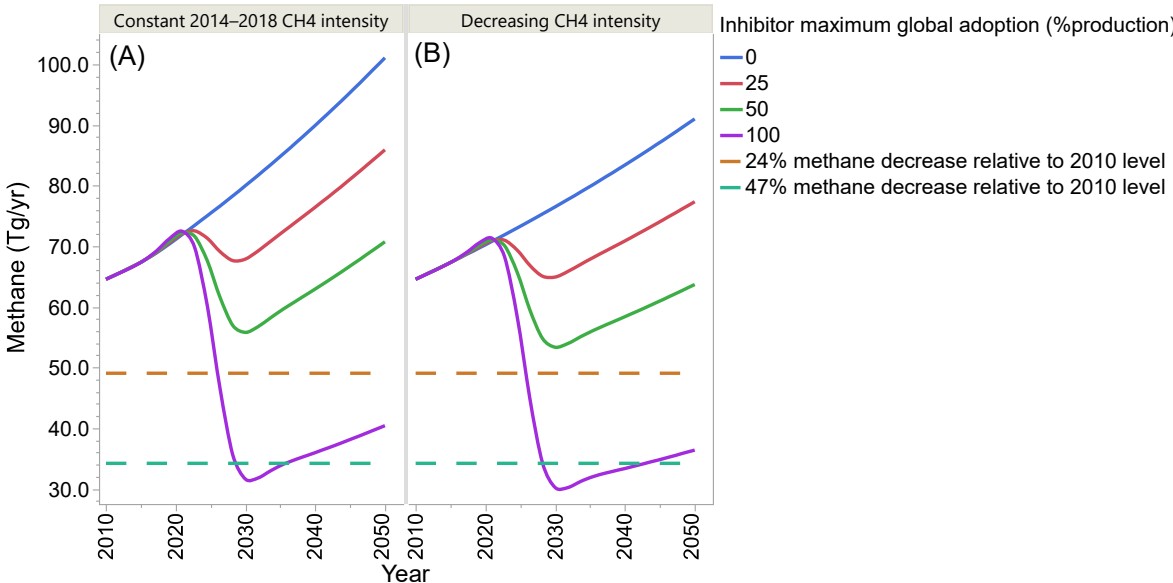

**Figure 8.** Predicted evolution of enteric methane ($CH_4$) emissions from global beef, lamb, and bovine milk production between 2010 and 2050 considering (**A**) constant $CH_4$ intensity and (**B**) declining $CH_4$ intensity according to Table 1. Adoption of feed additive inhibitors of methanogenesis decreasing $CH_4$ intensity by 60% for beef, lamb, and milk production is simulated to occur at 0, 25, 50, or 100% of global production. Decreases in enteric $CH_4$ emissions of 24 and 47% relative to 2010 required to maintain a global temperature increase within 1.5 °C according to different socioeconomic scenarios and climate models [15] are shown in dashed lines.

## 6. Cost Effectiveness and Co-Benefits of Inhibiting Methanogenesis

Successful adoption of antimethanogenic strategies requires that they are economically attractive to producers [50]. The inclusion of 3-NOP, *Asparagopsis*, or any other inhibitor of methanogenesis developed in the future in ruminant diets would raise feed costs, and if everything else remained unchanged, the use of antimethanogenic feed additives would be economically unattractive. Some possible means to overcome the added costs of antimethanogenic feed additives are:

1.  Economic incentives;
2.  Methanogenesis inhibition increasing feed efficiency;
3.  Adjusting basal diet composition to the inhibition of methanogenesis.

### 6.1. Economic Incentives

It could be possible to overcome an increase in feed costs resulting from the inclusion of antimethanogenic feed additives in ruminant diets by establishing premium prices for meat and milk from animals fed inhibitors of methanogenesis and emitting less $CH_4$. Ultimately, consumers would be paying a higher price for environmentally friendly labeled meat or milk produced with lesser emissions of GHG. However, the relative size of these types of niche markets at a global scale is uncertain [51]. Markets for environmentally friendly labeled products were estimated as being of limited size in Europe [52]. Most of the growth in the production of animal products is expected to occur in the developing world [13], where niche markets for food products are generally marginal. The payment of a premium price for meat or milk products to encourage the adoption of inhibitors of methanogenesis may be feasible only in a relatively minor proportion of ruminant production markets situated in developed economies. Furthermore, higher prices of meat and milk associated with lower enteric $CH_4$ and $CO_2e$ emissions might increase total $CO_2e$ by further stimulating production in countries that already have a relatively high consumption of animal products.

Methane taxes are also a possibility to implement to stimulate the adoption of antimethanogenic measures, including feed additives inhibiting methanogenesis. A rising global tax has been proposed as an effective measure to mitigate emissions of $CH_4$. Models have predicted that responses in anthropogenic $CH_4$ abatement resulting from taxation would be largest in the fossil fuels sector, followed by agriculture and, lastly, the waste sector; in the ranking of modeled responses to $CH_4$ taxation by subsectors, enteric fermentation is second to coal [2,53]. The same as with the premium prices directed towards lowering enteric $CH_4$ emissions, it may be questionable whether enteric $CH_4$ taxation schemes would be implemented in developing economies increasing their production of ruminant products, at least in the short and medium term.

*6.2. Methanogenesis Inhibition Increasing Feed Efficiency*

Since at least the 1930s, ruminant nutritionists have known that $CH_4$ formation in the rumen is a loss of energy for the host animal and thus an inefficient process for ruminant production: "In the ruminant the waste products of digestion include the unused feed residues in the feces and relatively large amounts of methane gas, which is formed as a result of fermentation and serves no useful purpose" [54]. The realization of $CH_4$ formation in the rumen as an energy loss motivated in the 1960s and in following decades various studies seeking to inhibit methanogenesis in rumen fermentation to improve the efficiency of ruminant production. Blaxter and Czerkawski [55] proposed that "The almost saprophytic role which methanogenic organisms appear to play in the rumen, obtaining their energy from the end products of the microbial fermentation of carbohydrates and amino-acids, and producing waste products of methane and heat in large amounts, suggests very strongly that if their activity could be reduced without an impairment of cellulolysis, an increase in the productivity of ruminants could be obtained". It was not until the 1980s and 1990s, when the concern about livestock $CH_4$ emissions causing global warming [56,57] was raised, that the main goal of the research on the inhibition of rumen methanogenesis started to shift from improving energy efficiency to decreasing the environmental impacts of ruminant livestock e.g., Johnson and Johnson [58],McCrabb et al. [59].

Because ruminants lose between 2 and 12% of ingested gross energy as $CH_4$ [58], the possibility of enhancing feed efficiency through conserving energy that is otherwise lost as $CH_4$ could be a strong incentive to pay for the utilization of feed additives to inhibit rumen methanogenesis. However, examination of the evidence does not lead to conclusions about a consistent improvement in feed efficiency or animal performance when rumen methanogenesis is inhibited ([60]; Table 1, Table 2 and Table S1).

It has been suggested that the moderate inhibition of $CH_4$ production observed in most studies may not translate into large enough gains of net energy to elicit significant differences in feed efficiency or animal productivity [17,18]. Experiments with a much larger number of animals would thus be needed to detect significant differences in feed efficiency; in that regard, Alemu et al. [61], working with 4048 steers in a commercial feedlot, reported a tendency towards a 2.5% increase in feed efficiency when $CH_4$ production was moderately inhibited by 26%. It seems reasonable to think that energy savings resulting from moderate decreases in $CH_4$ production may not be detected in most experiments with a much smaller number of animals and lower statistical power. However, experiments with pronounced decreases in $CH_4$ production, still do not show consistent improvement in feed efficiency or animal performance (Table 1, Table 2 and Table S1).

Another factor that may contribute to explaining why inhibiting rumen methanogenesis does not consistently benefit animal productivity is that not all of the energy spared from $CH_4$ formation is incorporated into products nutritionally useful for the ruminant host animal. Most notably, inhibiting $CH_4$ production typically results in the release of dihydrogen ($H_2$) [62], which, in the typical rumen fermentation with functional methanogenesis, is a fermentation intermediate found at low concentration and the main electron donor for $CH_4$ production [63]. Accumulation of $H_2$ is potentially problematic because $H_2$ expelled is, in itself, a loss of energy. In 14 experiments in which the inhibition of $CH_4$

production was greater than 50%, energy lost as $H_2$ followed a nonlinear and variable relationship with energy lost in $CH_4$. Energy losses as $H_2$ could be described as moderate. For example, energy losses in $H_2$ at 80% inhibition of methanogenesis accounted for 11% [$CI_{95}$ = 4.5, 17%] of the energy saved in $CH_4$ not formed. There has been speculation about incorporating $H_2$ into useful metabolic pathways through the use of electron acceptors or hydrogenotrophic microorganisms when methanogenesis is inhibited [64]. Benefits to animal productivity may not only depend on the sheer energy savings of $H_2$ incorporation, but also on the significance and metabolic fate of the absorbed sink of metabolic hydrogen for each type of animal, depending on its nutritional requirements [65].

Accumulation of $H_2$ resulting from inhibiting methanogenesis can impair the reoxidation of microbial NADH, increasing the $NADH/NAD^+$ ratio [66]. Insufficient availability of $NAD^+$ can theoretically halt fermentation [67]. Inhibiting methanogenesis in rumen microbial cultures induced $H_2$ accumulation and decreased fermentation as estimated from production of volatile fatty acids (VFA) [68]. In vivo, however, there is no conclusive evidence of negative effects of inhibiting methanogenesis on apparent digestibility or on VFA concentration adjusted by changes in DMI. It should be noted, however, that VFA concentration does not necessarily reflect VFA production. Changes in VFA production might be compensated by changes in rates of VFA absorption or passage, incorporation into microbial biomass, and changes in rumen volume; effects of inhibiting methanogenesis on the actual production of VFA have not been determined [17,60].

Apart from $H_2$, inhibition of methanogenesis in vitro [69] and in vivo e.g., Martinez-Fernandez et al. [70],Martinez-Fernandez et al. [71],Melgar et al. [72] has also resulted in increased concentrations of other electron carriers intermediate in rumen fermentation: formate, lactate, and ethanol. Lactate formed in propionate absorption through the rumen wall is used as a substrate for gluconeogenesis [73]. Direct absorption of lactate from the rumen seems to be influenced by adaptation to high concentrate diets [74]. Whether lactate could be absorbed from the rumen and utilized for gluconeogenesis in methanogenesis-inhibited animals is unknown. However, increases observed in lactate concentration in methanogenesis-inhibited rumens are much lower than what is observed in acidotic rumens and, if absorbed, it may likely have a relatively small influence on the ruminant's energy budget.

Results about the responses of formate and ethanol to methanogenesis inhibition are scarce. Formate can reach about 6–12 mM concentration in methanogenesis-inhibited rumens [70,71]; in other studies, it accumulated to a much lower concentration below 1 mM [72]. Ethanol concentration has been found to increase with methanogenesis inhibition, yet to relatively low levels [72,75]. It is unknown how the kinetics of formation and disappearance of formate and ethanol respond to pronounced methanogenesis inhibition and their metabolic fate, so as to establish the importance of these metabolites in the flow of carbon and metabolic hydrogen in the methanogenesis-inhibited fermentation and, fundamentally, their significance for the animal's energy budget.

### 6.3. Adjusting Basal Diet Composition to the Inhibition of Methanogenesis

Inhibiting methanogenesis is not an isolated intervention and causes profound changes in rumen fermentation. Alterations in both catabolic and anabolic processes may result in increased absorption of some nutrients and may open opportunities to decrease the need for them to be supplied by the basal diet.

Inhibiting rumen methanogenesis decreases the acetate-to-propionate concentration ratio, as predicted by the elevation of $H_2$ concentration [76] and confirmed in a meta-analysis of in vivo experiments [17]. Although propionate concentration adjusted by DMI did not respond to methanogenesis inhibition [17], it is still possible that, if increases in propionate production occur, they may have gone unnoticed when measuring propionate concentration because of compensatory changes occurring in propionate absorption. It is important to understand the responses of propionate production and absorption to methanogenesis inhibition because propionate is the main substrate for gluconeogenesis in ruminants [77].

If a positive response in propionate production to inhibiting methanogenesis could be shown experimentally through the use of labeled propionate, it would be important to understand the fate of the extra propionate absorbed and its metabolic consequences.

The meta-analysis by Loncke et al. [78] found that the formation of glucose from the sum of propionate, amino acids, and lactate increased at decreasing rates as their flow to the liver increased. This response suggests that glucogenic precursors may exceed the animal's demands for glucose as their availability increases. If inhibiting methanogenesis can be shown to increase propionate production in the rumen, perhaps basal diets could be modified to include less concentrates as glucogenic precursors and still match the animal requirements for glucose. This could allow decreasing feed costs in regions where concentrates are expensive. It might also prevent the decrease in DMI observed when $CH_4$ production is inhibited if the drop in DMI observed when inhibiting rumen methanogenesis [17,60] is caused by greater propionate oxidation in the liver acting as a satiety signal [79]. In regions of the world where cereal grains are not used for feeding ruminants, inhibiting methanogenesis might allow gluconeogenesis to be enhanced with forage-only diets.

Accumulated $H_2$ resulting from inhibiting rumen methanogenesis can also be incorporated into reductive acetogenesis, the reduction of $CO_2$ with $H_2$ to acetate. The addition of reductive acetogens to methanogenesis-inhibited in vitro fermentation was successful at incorporating accumulated $H_2$ into acetate formation [80–82]. Raju [83] showed the occurrence of reductive acetogenesis in sheep rumens and its increase when methanogenesis was inhibited by acetylene. Because of the higher $H_2$ threshold of reductive acetogens compared to the methanogens so far cultivated, it is expected that reductive acetogens could decrease $H_2$ accumulation, but $H_2$ concentration may still be higher compared to rumen fermentation with functional methanogenesis [64]. Animal production implications of enhancing reductive acetogenesis as a sink of metabolic hydrogen have been discussed [65].

The consequences of inhibiting rumen methanogenesis on microbial anabolism have received little attention. The incorporation of ammonium into the synthesis of microbial amino acids was stimulated by the methanogenesis inhibitor 9,10-anthraquinone in rumen cultures growing on starch but not on cellulose [84]. If this finding could be confirmed in vivo and with a broader range of inhibitors of methanogenesis and real diets fed to ruminants, it may be possible to replace greater proportions of expensive plant protein supplements with urea, again lowering feed costs and favoring cost effectiveness of the use of inhibitors of methanogenesis.

The effects of inhibiting methanogenesis on the rumen metabolism of fatty acids have potential implications for the quality of ruminant products. Decreases in milk fat percentage of vaccenic and rumenic acids, and mono- and polyunsaturated fatty acids observed when inhibiting rumen methanogenesis [72,85–89] suggest an increase in biohydrogenation, perhaps stimulated by the increased availability of reduced cofactors. This would be an undesirable consequence of the methanogenesis inhibition intervention, as the fatty acids profile in ruminant products would be richer in saturated fatty acids. Perhaps the decrease in mono- and polyunsaturated fatty acids could be lessened by adding to the diet sources rich in linolenic acid, such as fresh forages or linseed, when inhibiting rumen methanogenesis.

## 7. Adoption of Inhibitors of Methanogenesis in Grazing Systems

Globally, 37.4% of total enteric $CH_4$ emissions are generated by ruminants on free-ranging systems on rangelands and grasslands, 60.5% in mixed systems, and only 2.10% from beef cattle in feedlots [36]. Mixed systems can, in turn, comprise an ample range of production system typologies, from low-cost systems based on different proportions of pastures, concentrates, and agricultural and agro-industrial by-products to intensive dairy production operations with confined animals fed total mixed diets based on conserved forages and concentrates (as no dairy animals are classified in the Feedlot category in the FAO GLEAM database [36], it is understood that dairy cows consuming mixed diets in

intensive operations are classified in the Mixed systems category). Assuming that most intensive dairy operations using total mixed rations are located in North America and Europe, it can be estimated using data from the FAO GLEAM database [36] that about 7% of enteric $CH_4$ globally could be emitted from ruminants in intensive production systems with confined animals (i.e., feedlots and confined dairies; calculations not shown), the rest corresponding to extensive ranging systems and animals on pastures or crop residues, sometimes supplemented with concentrates or by-products.

For every peer-reviewed published study on enteric $CH_4$ mitigation conducted with grazing animals, 5.6 were conducted with confined animals (calculated from results by Vargas et al. [90]). Therefore, investment in research and development in the mitigation of enteric $CH_4$ production under intensive production conditions appears to be over-represented relatively to the contribution of confined production systems to global enteric $CH_4$ emissions and, conversely, information is lacking on enteric $CH_4$ mitigation in grazing systems. In particular, very little research has been conducted in extensive systems without any supplementation. This is especially important because, as discussed, antimethanogenic feed additives are the most potent means of decreasing enteric $CH_4$ emissions from ruminants [17,26,27,91] and they have been developed and evaluated to be delivered in feed supplemented to animals. Other means of delivery of antimethanogenic feed additives would have to be developed for extensively ranging animals without feed supplementation, such as salt and molasses lick blocks [92], boluses, or in drinking water [17]. Moreover, perhaps genes encoding for bromoform biosynthesis in *Asparagopsis* spp. could be genetically engineered in forages or directly in ruminants to deliver bromoform to the rumen in saliva or in rumen microbes. The latter possibilities, however, would require enough bromide content in the soil and in ingested forages, may have environmental implications, may affect the fitness and performance of bromoform-synthesizing plants, animals, or microbes, and could be both technically and economically difficult.

Other strategies to mitigate enteric $CH_4$ production in extensively ranging animals not receiving supplementation are being investigated, such as the selection of grazing animals producing less $CH_4$ [93], early life interventions with potential long-lasting effects [46,94], and immunization against rumen methanogens [95]. In general, mitigation of enteric $CH_4$ using these strategies has been mild or moderate, results have sometimes been contradictory, and more research is needed on their implications for animal productivity in different production systems. However, it is of much interest to continue conducting research in these antimethanogenic strategies which have the potential to be applied in extensive grazing systems to decrease the daily emissions of $CH_4$ per animal.

## 8. Safety and Other Aspects Important for the Adoption of Inhibitors of Methanogenesis

Adoption of feed additive inhibitors of methanogenesis towards pronounced mitigation of enteric $CH_4$ requires fulfilling various other aspects apart from effectiveness and persistency. Feed additives should not have negative effects on animal productivity and welfare, be safe for animals, consumers, and the environment, the decreases in enteric $CH_4$ emissions should not be compensated by upstream or downstream emissions of other GHG, additives must be possible to implement in the production systems being considered, must be approved by government agencies, be acceptable to consumers, and be economically attractive for producers to adopt.

3-Nitrooxypropanol is regarded as safe within recommended and experimentally evaluated doses [96,97]. Increasing the dose of 3-NOP would minimally increase upstream emissions of fossil fuel $CO_2$ associated with manufacturing and transporting 3-NOP, as, because of 3-NOP low levels of inclusion in diets, fossil fuel $CO_2$ associated with manufacturing and transporting 3-NOP is small in terms of $CO_2$e compared to $CO_2$e not emitted as $CH_4$ (calculations not shown).

Supplementing high doses of *Asparagopsis* can decrease DMI and milk production [49,89] and cause rumen mucosa abnormalities and inflammation [98,99]. Conversely, in other studies, supplementing *Asparagopsis* improved growth and feed efficiency [43,44]. Bro-

moform, the main $CH_4$-suppressing compound in *Asparagopsis*, is a suspected carcinogen and stratospheric-ozone-depleting agent [100], although the potential global ozone depletion caused by hypothetical global adoption of *Asparagopsis* was estimated to be relatively small [101]. Supplementing *Asparagopsis* has not resulted in the passage of bromoform to meat, milk, organs, or feces [43,44,49,89,98], with the exception of the first experimental day with non-adapted cows in the study by Muizelaar et al. [99]. However, supplementing *Asparagopsis* resulted in the passage of iodine and bromide to milk [89] and iodine to meat [44]. Bromoform is rapidly degraded by rumen cultures, mainly to dibromomethane [102], which is considered less toxic than bromoform [103]. An alternative might be the use of pure bromoform or dibromomethane (perhaps stabilized in a delivery complex, as it has been previously conducted with bromochloromethane [59]), which would allow the exact dosing of the active compound, independently of the content of bromoform in *Asparagopsis*. In addition, dosing pure bromoform (or dibromomethane) would avoid potential problems of excess iodine passing to milk and meat, although it would still result in bromide accumulation in milk.

## 9. Possibilities for Enhancing the Effectiveness of Inhibitors of Methanogenesis

Maximizing the efficacy of inhibitors of rumen methanogenesis, i.e., 60% or more, will be important to substantially mitigate enteric $CH_4$ emissions. Meta-analyses and individual studies show that the extent of inhibition of methanogenesis by 3-NOP [29,30,32] and *Asparagopsis* is positively related to their dose [43,44,49,89,98] and to the content of bromoform in the case of *Asparagopsis* [100]. Therefore, if high-end doses are used, the extent of inhibition of rumen methanogenesis by feed additives is potentially greater than the averages obtained in meta-analyses.

It must be noted, however, that responses to the dose of 3-NOP have not been linear in all individual studies evaluating multiple doses of 3-NOP [45,86,88,104] and it is possible that, under some conditions, the inhibition of methanogenesis may plateau at doses lower than maximal. It is important to understand the reasons, other than differing maximal doses examined in each study, behind the variation among studies in the linearity and the magnitude of the response of $CH_4$ decrease to the dose of the inhibitors. Aspects such as diet and animal, influencing the composition of the methanogenic community, may affect the magnitude of the responses to inhibitors of methanogenesis. For example, the response of $CH_4$ decrease to 3-NOP in beef and dairy animals has been shown to decline with increasing dietary NDF [29]. Conversely, using a very high dose of 1200 mg 3-NOP/kg of substrate DM in semicontinuous culture, Schilde et al. [105] decreased $CH_4$ production by 97% without observing interactions between the dose of 3-NOP and the percentage of concentrate in the substrate incubated; their results show that a high dose of 3-NOP overcame the expected lower inhibition with higher NDF.

Increasing the effectiveness of methanogenesis inhibitors can represent an avenue to decrease their cost, i.e., achieving more pronounced inhibition with current average doses or the same extent of inhibition with lower doses than current averages. Understanding the effects of 3-NOP on different methanogens [106,107], as well as elucidating the mechanisms that contribute to the resistance of methanogens to inhibitors [108], can help design means to improve their efficacy and cost effectiveness.

Differential sensitivity among different methanogens grown in pure culture to 3-NOP [109] and to the chemical inhibitor of methanogenesis 2-bromoethanesulfonate (BES) [83,110] has been reported before. Both 3-NOP [109] and BES [111] inhibit methanogenesis as structural analogs of methyl-coenzyme-M, a methylated cofactor involved in the last step of methanogenesis. Whilst inhibition caused by 3-NOP is persistent [41,42,86], inhibition caused by BES in sheep lasted for only 3 d, after which methane production returned to pretreatment levels [112].

*Methanobrevibacter ruminantium* M1, which has lost three genes required to synthesize coenzyme M [113] and, therefore, requires coenzyme M included in its growth medium [114], took up coenzyme M from the medium with a high-affinity transport system. Conversely, *Mbr. ruminantium* PS, which synthesizes coenzyme M, took up coen-

zyme M with a rate of less than 10% compared to *Mbr. ruminantium* M1 [115]. Mutants of *Methanococcus voltae* resistant to BES had a considerably reduced capacity to transport BES into the cell [116]. Inhibition of methane production by BES in *Mbr. ruminantium* M1 and *Methanosarcina* spp. could be diminished or reversed by the addition of coenzyme M to the medium, demonstrating a competition for transport between coenzyme M and BES [115,117]. The same as with BES, it appears conceivable that differences among methanogens in sensitivity to 3-NOP could also be related to the transport of 3-NOP into the cell and the ability of methanogen species to synthesize coenzyme M. It is possible that the effectiveness of 3-NOP at inhibiting methanogenesis is influenced, among other factors, by the proportion of coenzyme-M-synthesizing methanogens in the rumen methanogenic community and by the concentration of coenzyme M in rumen fluid.

In vivo work has also revealed shifts in the methanogenic community when inhibitors of methanogenesis are supplemented. Chloroform inhibited *Mbr. gottschalkii* more than *Mbr. ruminantium* [71], and both chloroform and 3-NOP were more inhibitory to hydrogenotrophic *Methanobrevibacter* spp. than to methylotrophic *Methanosphaera* spp. [75,106,118]. Those results agree with Duin et al. [109], who found that *M. ruminantium* was the most sensitive to 3-NOP of the methanogens they evaluated, *Msp. stadtmanae* was more resistant, and methylotrophic *Ms. barkeri* and hydrogenotrophic *Methanomicrobium mobile* were the most resistant. On the other hand, 3-NOP decreased the abundance of *Mmb. mobile* in rumen fluid [118]. Ungerfeld et al. [110] also found that methylotrophic *Ms. mazeii* was more resistant to BES and other inhibitors than *Mbr. ruminantium*, with *Mmb. mobile* being intermediate. It seems then that some chemical inhibitors evaluated may preferentially target hydrogenotrophic, over methylotrophic, methanogens. It is of much interest to understand if variation in the sensitivity of methanogens to particular chemical inhibitors is related to their metabolic pathway of $CH_4$ formation, i.e., hydrogenotrophic vs. methylotrophic methanogenesis, as the dietary content of methyl group precursors can influence the make-up of the methanogenic community and the relative importance of both methanogenic pathways. Furthermore, as $H_2$ thresholds differ among hydrogenotrophic, methyl-reducing, and methyl-fermenting methanogens [119], differential effects of 3-NOP on the different groups of methanogens could have implications for the extent of $H_2$ accumulation and release occurring as a consequence of inhibiting rumen methanogenesis.

Antimethanogenic compounds differ in their mechanisms of action. As discussed, 3-NOP [109] and BES [111] inhibit methyl-coenzyme M reductase through being structural analogs of methyl-coenzyme M, a cofactor present in all known methanogens. Methane halogenated analogs, such as chloroform or bromoform, react with cobamides and block the transfer of a methyl group from tetrahydromethanopterin to coenzyme M [120]. Derivatives of *p*-aminobenzoic acid inhibit the synthesis of methanogenic cofactor tetrahydromethanopterin [121], and statins inhibit methanogen growth by impairing membrane lipid synthesis [122]. Through the understanding of these mechanisms of action, it may be possible to design different combinations or rotations of antimethanogenic compounds to target specific methanogenic communities varying in composition depending on diet and animal. Because methanogens less inhibited by a particular compound may partially occupy the niche left by those methanogens inhibited more severely, it is conceivable that combining antimethanogenic compounds targeting different methanogens could result in synergic effects.

Apart from microbiological factors related to the sensitivity of different methanogens to inhibitors, the effectiveness of inhibitors of methanogenesis is also influenced by the daily pattern of inhibitor concentration in the rumen. Rumen concentration of inhibitors of methanogenesis is affected by the mode of administration, the time elapsed since the last feeding episode, the rates of feed ingestion and rumen fluid outflow, changes in rumen volume, and rates of metabolism and absorption of each specific compound. Almost all 3-NOP is metabolized to 1, 3-propanediol within 24 h and about 50% within 7 h [109]. Absorption of 3-NOP occurred in orally dosed rats, with plasma concentration peaking at 5–15 min after dosing [97], with no published results being available for ruminants. van

Lingen et al. [123] modeled a peak in rumen concentration of 3-NOP of 0.055 mM 1.5 h after feeding, which gradually fell to 0 mM at about 12 h after feeding in animals fed twice per day a diet with 121 mg 3-NOP/kg DM. Bromoform was degraded in rumen microbial cultures by 70 and 90% after 30 min and 3 h incubation, respectively [102].

Predicted fluctuations in 3-NOP concentration in the rumen agree with the diurnal pattern of $CH_4$ emissions of animals fed once [88] or twice a day [45,61]. Dosing 3-NOP through the rumen cannula resulted in a relatively strong but short inhibition of methanogenesis, ultimately resulting in less than a 10% decrease in daily methane production, presumably because of the rapid washout of 3-NOP from the rumen [124]. Delivering methanogenesis inhibitors into the rumen in slow-release forms may result in more sustained concentration and greater effectiveness at inhibiting $CH_4$ production, even in animals fed total mixed rations. On the other hand, slow-release forms would likely increase manufacturing costs.

## 10. Final Remarks

Enteric $CH_4$ emissions from ruminants are a moving target for mitigation. This is because, as ruminant production increases, the decrease in total enteric $CH_4$ emissions necessary to contain global warming augments with time. A conclusion of simulations of projected enteric $CH_4$ emissions under constant or decreasing $CH_4$ intensity, and different extents of adoption and effectiveness of inhibitors of methanogenesis, was that antimethanogenic feed additives would have to be adopted at very large rates worldwide to decrease enteric $CH_4$ emissions as needed to limit the global temperature increase to 1.5 °C. In addition, antimethanogenic feed additives would have to consistently attain pronounced inhibition of rumen methanogenesis, which has been shown to be possible, although is beyond the average inhibition obtained in most studies. Inhibitors of methanogenesis are the single most potent enteric $CH_4$-mitigation strategy, but they will have to be complemented with continuous improvements in production efficiency and other $CH_4$ mitigation strategies. Carbon sequestration in soil can also be important to offset emissions of GHG, especially in degraded soils, but it should be taken into account that carbon accretion in soil does not proceed indefinitely. The side of the equation which has been assumed as fixed in this analysis, that is, the demand for ruminant products, should also be critically examined, bearing in mind the multiple nutritional, economic, social, and environmental implications of ruminant production. Reduction in the consumption of ruminant products may be considered in regions where it exceeds the nutritional requirements of human populations, as well as reductions in the use of arable land and human edible food in ruminant production and in food waste.

The use of antimethanogenic feed additives will increase feeding costs, which, everything else being the same, will discourage their adoption. This is especially true for extensive production systems that rely solely on grazing or on the use of low-cost by-products. For commercial extensive production systems, such as ranching, keeping low costs of production is key to remaining competitive in the business, whereas the possibilities for smallholders and pastoralists to increase production costs are limited by lack of finance and technology, access to markets, volatility of prices, and risks to sheer subsistence. Moreover, in intensive and semi-extensive operations, the adoption of antimethanogenic feed additives would likely be unattractive unless coupled with some benefit.

A premium price offered for meat and milk associated with lower $CH_4$ emissions could be attractive to producers so as to encourage the use of antimethanogenic feed additives. Niche markets ready to pay a higher price for environmentally produced livestock products are conceivable in developed economies but less likely in developing countries, where most of the growth in animal production is forecasted to take place. Methane taxes can be another economic incentive for $CH_4$ mitigation but, again, they may be more feasible to implement in developed economies, at least initially.

Research on understanding the alterations in the rumen and whole animal metabolism and physiology caused by inhibiting rumen methanogenesis may unveil new avenues for improving feed efficiency or reformulating basal diets and lowering their cost. This path,

however, is relatively long-term, and its results are uncertain. Continued efforts in this direction are important but should also be accompanied by the implementation of proven antimethanogenic strategies causing moderate or mild decreases in emissions of enteric $CH_4$ and $CO_2e$, which can, at the same time, be productively and economically attractive.

There are important nutritional, economic, social, and environmental roles of ruminant production, its integration with crop production, and its ability to use nonarable land. The demand for ruminant meat and milk is increasing, especially in developing economies, thus also increasing the emissions of $CH_4$ from ruminant production. On the other hand, due to the relatively short half-life of $CH_4$ in the atmosphere, rapid action to decrease the emissions of enteric $CH_4$ from ruminant production offers an opportunity for short-term positive impact on climate change. It is necessary to investigate, develop, and adapt solutions, including antimethanogenic compounds, to the wide diversity of ruminant production systems worldwide.

**Supplementary Materials:** The following supporting information can be downloaded at: https://www.mdpi.com/article/10.3390/methane1040021/s1, Table S1: In vivo experimental treatments resulting in a 60% or greater decrease in enteric methane ($CH_4$) production.

**Funding:** This research was funded by Agencia Nacional de Investigación y Desarrollo ANID through project Fondecyt 1190574.

**Institutional Review Board Statement:** Not applicable.

**Informed Consent Statement:** Not applicable.

**Data Availability Statement:** Publicly available datasets were constructed by the author for this study. Data were made publicly available by the author on August 30, 2022, and can be found at: Open Science Framework: https://osf.io/drste/?view_only=2d0ee909617444a8b8568e50721d6e01, accessed on 30 August 2022.

**Conflicts of Interest:** The author declares no conflict of interest. The funder had no role in the design of the study; in the collection, analysis, or interpretation of data; in the writing of the manuscript; or in the decision to publish the results.

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
