# Peer review of "Opportunities and Hurdles to the Adoption and Enhanced Efficacy of Feed Additives towards Pronounced Mitigation of Enteric Methane Emissions from Ruminant Livestock"

_methane, doi:10.3390/methane1040021_

Round 1

Reviewer 1 Report

General comments

I committed to review this manuscript based on the statements in the abstract. I thought it would be a very important contribution to the knowledge addressing the emissions of enteric methane from livestock. However, in the process of reading I found that the simulation is highly theoretical with not much of substance to inform research in the short- or longer-term. The practical application of outcomes of the work is far away from realisation. This relates to the fact that only around 2% of the global enteric CH4 emission are associated with intensive production systems, yet the modelling exercise addresses the mitigation impact on production volume basis rather than animal populations and locations. This fact is clearly stated in the Section 8. The likelihood of adoption of the feed additives in the systems where most of the animals are found would be minimal, especially when there will be no co-benefits, especially net profit. If these facts were stated earlier in the manuscript, I would have rejected the MS earlier in the review. The manuscript fails in discussing the effect of other pathways of mitigating emissions; whether additive effects would be possible. Overall, the manuscript is well written.

Specific comments/suggestions

Lines 2-4. … Title do not reflect objectives. Does not mention the factor related to emission intensity (is this synonymous of sustainable intensification)?. Insert …emissions from livestock

Line 39. How relevant is crop yield to this manuscript? perhaps forage primary production?

Lines 40-41. Is this independent from reduction in other GHG?

Lines 41-43. Not clear

Lines 56-67. Intensification is heavily constrained by environmental (abiotic) conditions in most livestock systems/typologies.

Line 78. Increase in animal numbers and animal body size?

Lines 84-85. Repeating 24 and 47%

Line 102. Animal production vs animal population

Lines 103-107. Intensification will not be cost-effective, neither sustainable in systems under high environmental constraints. On the contrary it can have undesired consequences.

Line 102. Most of livestock most likely are already being fed at full rumen capacity (with bulky and low digestible feeds). So, 58% increase is unrealistic.

Line 128. What about safety for the animals and the consumers?

Line 130-135. Is this realistic? Some academic seems are unaware of global livestock systems.

Line 159. Production of what?

Lines 173-178. What is the driver for such an adoption?

Line 184..production. For meat

Line 193. Given that Chang

Lines 269-271. So what is the purpose of this manuscript? What other means are there to drive adoption?

Line 271. Four two

Table 2. DMI, ADG and G:F, all are worrying. There are negative effects or NS; then no driver for adoption

Lines 295 and 296. Liveweight gain not body mass gain

Line 363 ‘decrease’ use another word, avoid confusion

Line 365. Is this in vitro stuff relevant?

Line 368. Affecting… affect

Lines 370-378. Drivers for adoption, affordability, cost-effectiveness, delivery pathways, perishability ?

Lines 391-393. How long were these studies. In the longer term it may

Line 420. Uncertain or very small?

Lines 440-441. Not absolutely true. On forage diets of poor quality it is necessary for the process to occur. How important is the lost of energy?. 6-8% of GEI, how important is compared to loss in indigestibility for example?

Line 457. In practice, these extremes are not present.

Line 464. Saving of metabolizable energy? MEI=GEI-FE-UE-GE

Line 484. H2 is also called ‘intermediate GHG’ it enhances the lifetime of CH4 because consumes OH

Lines 649-670. It would be desirable to discuss ‘delivery’ in a separated sub-section

Line 671. This is the most important section to me, but sadly touched only towards the end

Line 673, ruminants in free ranging system on rangelands and grasslands

Line 674. Production systems/typologies

Line 675 extensive systems based only on the utilisation of natural grasslands/rangelands, crop-livestock mixed systems. I am sorry, you must describe better this whole paragraph. Do not confound. Pastures mean cultivated forages (annual or perennial, used mostly by grazing). TMR is for intensive systems (feedlots). Mixed systems may use concentrates to supplements forages or crop residues.

Lines 683-685. Most of animal research are done with few animals under highly constrained/controlled environment and for a short period: days or few weeks. Obviously are not representative of production systems.

Line 686. Precisely, all this knowledge are being extrapolated to extensive free-ranging situations. Most of researchers in LMIC copy that knowledge.

Line 712. I have seen hundreds of reviews and outputs of modelling that unfortunately miss the context of global livestock production, trying to sell the high-hanging fruits before the low hanging fruits are not being exploited. Here, for example, I do not see a discussion of the best animal husbandry practices impact on reduction of Ei as well as absolute emissions (culling unproductive animals, reducing herd maintenance feed intake, etc). Neither there is discussion of other pathways of emissions reduction.

Lied 726-728. Synergies with other desirable outcomes? Animal welfare, other environmental impacts?

Reviewer 2 Report

1.       Figures 1-4 should be re-designed into one figure.

2.       Why do the lines in between figure 1 (a) and (b) have no difference?

3.       Again, figures 5-8 should be re-designed into one figure.

4.       Please discuss more on the difference between plots (a) and (b) in each figure (figures 1-8). 

Reviewer 3 Report

Comments and Suggestions for Authors

The manuscript entitled “Opportunities and hurdles to the adoption and enhanced efficacy 2 of feed additives towards pronounced mitigation of enteric 3 methane emissions” has been carefully evaluated. The study attempts to determine the mitigation of enteric methane (CH4) emissions from ruminants with the use of feed additives inhibiting rumen methanogenesis to limit the global temperature increase to 1.5 °C.

Please note my specific comments and suggestions below:

Abstract

Line 10: Change “additives inhibiting of rumen methanogenesis to limit global temperature” to additives inhibiting rumen methanogenesis to limiting the global temperature …

Lines 13: Change “contribute to limit global temperature” to contribute to limiting global temperature...

Line 19: Change “cost effective” to cost-effective…

Line 20: Change “community, and rate” to community and the rate…

Line 26: Change “cost effectiveness” to cost-effectiveness…

Comment: It is convenient to include a chapter of “General introduction” that establishes the importance and justification of the subject to be analyzed in this document and establishes the main objective.

1. Enteric methane emissions and climate change

Line 34: Change “strong, rapid and sustained” to strong, rapid, and sustained…

Lines 36-37: Change “period) and relatively short life (9.25 ± 0.6 years) and perturbation” to  period) and relatively short life (9.25 ± 0.6 years), and perturbation…

Line 39: Change “due to ground level ozone” to due to ground-level ozone…

Line 42: Change “recent indicate continuous growth in” to recent indicates continued growth in…

Line 47: Change “by livestock have increased by” to by livestock increased by…

Line 50: Change “estimated to be of 20% [2]” to estimated to be 20% [2]…

Line 53: Change “inhibitors of methanogenesis, and” to inhibitors of methanogenesis and…

2. Intensification, productivity, and enteric methane emissions

Line 57: Change “production increases feed intake” to production increases feed the intake…

Line 82: Change “order to contain global temperature” to order to contain a global temperature…

Line 88: Change “beef, lamb and milk production” to beef, lamb, and milk production…

Line 91: Change decreasing global CH4 emissions intensity of beef, lamb and milk production” to decreasing the global CH4 emissions intensity of beef, lamb, and milk production…

Line 98: Change “of animal product has allowed to lower the total number” to of the animal product has allowed lowering the total number…

Lines 100-101: Change “other cases the decrease in the emissions of CO2e 100 per unit of animal product” to other cases, the decrease in the emissions of CO2e 100 per unit of the animal product

Line 106: Change “and much less to decrease it.” to and much less decrease it...

3. Mitigation of enteric methane emissions

Line 114: Change “management, addition of oils to” to management, the addition of oils to…

Line 123: Change “decrease CH4 intensity” to decrease the CH4 intensity…

4. Projection of global enteric methane emission under different scenarios of intensification and adoption of inhibitors of methanogenesis

Lines 142-143: Change “milk, and, dependig on each scenario” to milk, and, depending on each scenario…

Line 147: Change “for the 2014-2018” to for 2014-2018…

Lines 157-158: Change “two, with an increasing CH4 intensity on average. Assuming that global CH4 intensity” to two, with increasing CH4 intensity on average. Assuming that the global CH4 intensity.

Lines 162-163: Change “to the 2000-2004 quinquennial) and 2016 (corresponding to the 2014-2018” to to 2000-2004 quinquennial) and 2016 (corresponding to 2014-2018 quinquennial …

Lines 179-182: Change “Methane intensity of beef, lamb, or bovine milk in each year and scenario was multiplied by global production of the corresponding animal product to obtain total CH4 emissions associated to each product. Initially, the use of inhibitors of methanogenesis in 181 milk production was modelled as causing” to The methane intensity of beef, lamb, or bovine milk in each year and the scenario was multiplied by the global production of the corresponding animal product to obtain the total CH4 emissions associated to each product. Initially, the use of inhibitors of methanogenesis in milk production was modeled as causing…

Line 187: Change “Stronger effect of” to A stronger effect of…

Line 190: Change “increases of production of” to increases in production of…

Lines 202-203: Change “targets of 24 and 47% decrease in enteric CH4 emissions by 2050 relative to 2010 levels, as 202 required to maintain global temperature” to targets of a 24 and 47% decrease in enteric CH4 emissions by 2050 relative to 2010 levels, as 202 required to maintain a global temperature…

Line 212: Change “maintain global temperature” to maintain a global temperature…

Line 220: Change “maintain global temperature” to maintain a global temperature…

Line 228: Change “maintain global temperature” to maintain a global temperature…

Lines 236-237:maintain global temperature” to maintain a global temperature…

Line 246: Change “was not be projected” to were not projected…

5. Pronounced inhibition of rumen methanogenesis with feed additives

Lines 261-262: Change “but animal performance” to more, but the animal performance…

Line 167: Change " noted that generally the” to noted that generally, the…

Lines 271-272: Change “in 60% of more in CH4” to of 60% or more in CH4…

Line 282: Change “with an efficacy of decreasing CH4” to with the efficacy of decreasing CH4…

6. Opportunities and hurdles to increase the adoption and efficacy of methanogenesis inhibitors

Line 285: Change “appears that, under the likely” to appears that under the likely…

Line 287: Change “maintain global temperature increase” to maintain a global temperature…

Line 301: Change “estimated from graph” to estimated from the graph…

Line 319: Change “maintain global temperature” to maintain a global temperature…

Line 327: Change “maintain global temperature” to maintain a global temperature…

Line 335: Change “Change “maintain global temperature” to maintain a global temperature…

Line 342: Change “beef, lamb and milk production” to beef, lamb, and milk production…

Line 344: Change “required to maintain global temperatura” to required to maintain a global temperature

Line 347: Change “extent of inhibition, the” to the extent of inhibition the…

Line 350: Change “Importantly however,” to Importantly, however,…

Line 354: Change “and content of” to and the content of…

Line 357: Change “that responses to dose of” to that response to the dose of…

Line 362: Change “production to the dose of inhibitors” to production to the dose of inhibitors…

Line 371: Change “requires of various aspects” to requires various aspects…

Line 373: Change “animal productivity, be safe for animals, consumers and the” to animal productivity, and be safe for animals, consumers, and the…

Line 382: Change “associated to” to associated with…

Lines 391-392: Change “resulted in passage of bromoform to meat, milk organs or feces” to resulted in the passage of bromoform to meat, milk organs, or feces…

7. Cost-effectiveness of pronounced mitigation of enteric methane emissions

7.1. Economic incentives

Line 416: Change “in ruminant diets through” to in ruminant diets by…

Lines 418-419: Change “friendly labelled meat or milk” to friendly labeled meat or milk…

Lines 420-421: Change “of these type of niche markets at a global scale is uncertain [65]. Most of the growth in production of animal” to of these types of niche markets at a global scale is uncertain [65]. Most of the growth in the production of animals…

Line 422: Change “of a premium prize for meat” to of a premium price for meat...

Line 434: Change “of modelled responses” to of modeled responses…

7.2. Methanogenesis inhibition increasing feed efficiency

Line 445: Change “and following decades various studies seeking to” to and in following decades various studies sought to…

Line 449: Change “and amino-acids, and” to and amino acids, and…

Line 459: Change “thought as a strong incentive” to the thought of as a strong…

Line 460: Change “does not lead to conclude” to does not lead to conclusions…

Lines 465-466: Change “Experiments with much larger number of” to Experiments with a much larger number of…

Line 471: Change “with much smaller number of” to experiments with a much smaller number of…

Line 474: Change “may contribute to explain why” to may contribute to explaining why…

Line 477: Change “results in release of dihydrogen” to typically results in the release of dihydrogen…

Lines 491-492: Change “can impair re-oxidation of” to can impair the re-oxidation of…

Lines 493-494: Change “methanogenesis in rumen in” to methanogenesis in the rumen in...

Line 501: Change “on actual” to on the actual…

Line 514: Change “about responses of” to about the responses of…

7.3. Adjusting basal diet composition to the inhibition of methanogenesis

Lines 526-527: Change “decrease the needs for them” to decrease the need for them…

Line 528: Change “decreases the acetate to propionate” to decreases the acetate-to-propionate…

Lines 529-530: Change “in meta-analysis of” to in a meta-analysis of…

Line 537: Change “the use of labelled propionate” to the use of labeled propionate…

Line 543: Change “their availability increase” to their availability increases…

Line 548: Change “is inhibited, if” to is inhibited if

Line 550: Change “as satiety signal” to as a satiety signal…

Line 551: Change “might allow enhancing” to might allow for enhancing…

Line 555: Change “reductive acetogens were” to reductive acetogenins were…

Line 556: Change “fermentation were successful” to fermentation was successful…

Line 559: Change “higher H2-threshold” to higher H2 threshold…

Line 563: Change “reductive acetogenesis as” to reductive acetogenins as…

Line 564: Change “Consequences of” to The consequences of…

Line 565: Change “Incorporation of” to The incorporation of…

Line 570: Change “cost effectiveness of” to cost-effectiveness of…

Line 572: Change “on rumen metabolism of” to on the rumen metabolism of…

Line 576: Change “stimulated by increased availability” to stimulated by the increased availability…

Line 581: Change “linseed, when inhibiting” to linseed when inhibiting…

7.4. Improving the effectiveness of inhibitors of methanogenesis

Line 584: Change “current average doses, or” to current average doses or…

Line 589: Change “can help designing means” to can help design means…

Line 590: Change “cost effectiveness” to cost-effectiveness…

Line 594: Change “as structural analogues of” to as structural analogs of…

Line 605: Change “by addition of“ to by the addition of…

Line 608: Change “related to transport of” to related to the transport of…

Lines 635-636: Change “reductase through being structural analogues of” to reductase by being structural analogs of…

Line 637: Change “Methane halogenated analogues such” to Methane-halogenated analogs such…

Line 658: Change “modelled” to modeled…

Line 661: Change “in rumen in” to in the rumen in…

Line 665: Change “in less than 10% decrease” to in less than a 10% decrease…

Line 666: Change “because of rapid washout of” to because of the rapid washout of…

Line 667: Change “slow release forms” to slow-release forms…

Line 669: Change “slow release forms” to slow-release forms…

 8. Adoption of inhibitors of methanogenesis in extensive systems

Lines 676-677: Change “and concentrates used” to and concentrate used…

Line 684: Change “calculated from” to calculated by…

Line 696: Change “slow release” to slow-release…

9. Final remarks

Line 713: Change “from ruminants is a moving” to from ruminants are a moving…

Line 717: Change “inhibitors of methanogenesis, yielded” to inhibitors of methanogenesis yielded…

Line 718: Change “compatible to the” to compatible with the…

Line 720: Change “adopted worldwide, and” to adopted worldwide and…

Line 729: Change “Use of” to The use of…

Line 731: Change “low cost by-products to low-cost by-products…

Line 733: Change “key to remain competitive” to key to remaining competitive…

Line 738: Change “coupled to some benefit” to coupled with some benefit…

Lines 742-743: Change “livestock products is conceivable in developed economies, but may be less” to livestock products are conceivable in developed economies, but may be less…

Line 744: Change “contribute to catalyze the” to contribute to catalyzing the…

Line 746: Change “high income markets ready to”high-income markets are ready to…

Line 749: Change “may be”to maybe…

Line 751: Change “in rumen” to in the rumen…

Line 753: Change “efficiency or lower the cost” to efficiency or lowering the cost…

Line 756: Change “towards enhancing adoption” to towards enhancing the adoption…

Line 758: Change “well as implementation proven antimethanogenic” to well as implementation proved antimethanogenic…

Line 774: Change “allow designing antimethanogenic” to allow the designing antimethanogenic…

Line 778: Change “for pronounced mitigation of enteric” to for the pronounced mitigation of enteric…

Line 780: Change “lead to critically examine the” to lead to critically examining the…

Line 782: Change “is a complex aspect, because” to is a complex aspect because…

Lines 787-788: Change “its ability to use of non-arable land. Reduction in the consumption of ruminant products where it exceeds the” to its ability to use non-arable land. Reduction in the consumption of ruminant products which exceed the…

Line 789: Change “populations, in the use of” to populations, the use of”

Line 790: Change “and of food waste, are” to and food waste, are…

Line 794: Change “resulting in 60% or more” to resulting in a 60% or more…

Line 801: Change “This data can be found at” to Data can be found at…

Line 803: Change “The author declare no conflict of interest” to Authors declare no conflict of interest…

Line 804: Change “analyses, or” to analysis, or

Comments. It is advisable that the document contains a chapter on Conclusions. Generally, conclusions contain the most relevant findings of the research, are written in long lines, do not contain citations or references, and do not repeat results verbatim.
